# Direct and selective access to amino-poly (phenylene vinylenes)s with switchable properties by dimerizing polymerization of aminoaryl carbenes

Quentin Sobczak[1,2], Aravindu Kunche[1,2], Damien Magis [2], Daiann Sosa Carrizo [3], Karinne Miqueu[3], Jean-Marc Sotiropoulos [3], Eric Cloutet[1], Cyril Brochon[1], Yannick Landais [2], Daniel Taton [1✉] & Joan Vignolle [1✉]

Despite the ubiquity of singlet carbenes in chemistry, their utility as true monomeric building blocks for the synthesis of functional organic polymers has been underexplored. In this work, we exploit the capability of purposely designed mono- and bis-acyclic amino(aryl)carbenes to selectively dimerize as a general strategy to access diaminoalkenes and hitherto unknown amino-containing poly(p-phenylene vinylene)s (N-PPV's). The unique selectivity of the dimerization of singlet amino(aryl)carbenes, relative to putative C-H insertion pathways, is rationalized by DFT calculations. Of particular interest, unlike classical PPV's, the presence of amino groups in α-position of C=C double bonds in N-PPV's allows their physico-chemical properties to be manipulated in different ways by a simple protonation reaction. Hence, depending on the nature of the amino group ($iPr_2N$ *vs.* piperidine), either a complete loss of conjugation or a blue-shift of the maximum of absorption is observed, as a result of the protonation at different sites (nitrogen *vs.* carbon). Overall, this study highlights that singlet bis-amino(aryl)carbenes hold great promise to access functional polymeric materials with switchable properties, through a proper selection of their substitution pattern.

[1] Laboratoire de Chimie des Polymères Organiques, CNRS, Université de Bordeaux IPB-ENSCPB, Pessac Cedex, France. [2] Université de Bordeaux, ISM, UMR 5255, 33400 Talence, France - CNRS, ISM, UMR 5255, Talence, France. [3] CNRS, Université de Pau & Pays de l'Adour E2S UPPA, IPREM UMR 5254, Hélioparc., Pau cedex 09, France. ✉email: taton@enscbp.fr; vignolle@enscbp.fr

Singlet carbenes have witnessed tremendous developments since the isolation of the first stable carbenes by Bertrand et al.[1] in 1988 and Arduengo et al.[2] in 1991. In particular, a great variety of stable singlet carbenes has been prepared over the past 30 years[3–8]. Most examples reported to date feature a cyclic backbone and are thermodynamically stabilized thanks to the presence of one or two amino groups next to the carbene center, as highlighted with cyclic (alkyl)(amino)carbenes[9] and N-heterocyclic carbenes[10], respectively. Nevertheless, some acyclic mono-aminocarbenes (MACs) have also been isolated[11,12], providing that the carbene center benefits from sufficient steric protection. Indeed, in the absence of sufficient kinetic stabilization, dimerization of the carbene into the corresponding tetra-aminoethylene and diaminoethylene has been observed in the case of (cyclic) di-amino and (acyclic) mono-aminocarbenes, respectively[13–17]. Such a dimerization reaction has been proposed to occur following a non-least motion pathway, involving the attack of the occupied in-plane σ orbital of one carbene onto the out-of-plane $2p_\pi$ orbital of a second carbene[15,18–20]. In fact, dimerization, as well as the reverse process reforming the carbene, appear to be most generally catalyzed by electrophilic impurities[21–24].

While this dimerization reaction is often considered as deleterious for the isolation of carbenes, it eventually offers a powerful way to construct C=C double bonds. Surprisingly, in the context of polymer chemistry, dimerization of bis-carbenes has received little attention so far[25–28]. Bielawski et al. reported that some bis-benzimidazol-2-ylidenes, generated by deprotonation of the corresponding bis-benzimidazoliums, could undergo such a dimerization/polymerization, forming a class of thermally reversible dynamic covalent conjugated polymers[29–31], though showing a low conjugation across the aryl spacer[32] in addition to be rather air-sensitive.

Formal polymerization by dimerization of non-heteroatom-stabilized bis-(aryl)carbenes, generated by decomposition of corresponding bis-diazo compounds, has also been reported[33–36]. Depending on the nature of the catalyst, either poly(phenylene vinylene)s (PPV's) or poly(aryl azine)s could be achieved, upon extrusion or retention of $N_2$, respectively. PPV's represent an attractive class of highly stable, non-dynamic, and fluorescent polymeric semi-conductors that have received considerable attention[37,38]. Thus, besides the aforementioned diazo route, a variety of different synthetic approaches to PPV-like materials, involving either a step-growth or a chain-growth polymerization route, have been developed[39,40]. In general, related synthetic methods provide safer access to PPV's, in comparison to those involving the decomposition of diazo-containing precursors. They also allow the structural diversity of PPV's to be significantly broadened (in contrast, diazo precursors feature electron-withdrawing substituents only)[41]. While specific groups have been incorporated into the aromatic moiety of PPV's as a means to increase their solubility and facilitate their processability, introduction of heteroatoms directly linked to the double bond would yield PPV's with original optoelectronic properties[42,43]. These properties may even be finely tuned upon post-chemical modification of the basic amino groups[44] in 1,2-position of the double bonds, in contrast to the behavior of PPV's free of such α-amino substituents. We thus reasoned that these amino-PPV's, denoted as N-PPV's, i.e. containing two amino groups directly connected to the double bonds, could be accessible following a direct and selective dimerizing polymerization approach involving hitherto unknown bis-aminoaryl carbenes (see Fig. 1).

Here we report that deprotonation of purposely designed aldiminium salt precursors allows for the efficient in situ generation of acyclic amino(aryl)carbenes. The latter intermediate

species are then found to selectively and quantitatively dimerize into the corresponding diaminoalkenes. Such selectivity is rationalized by comparing the energy profiles of the dimerization process, relatively to putative competitive intramolecular C–H insertion pathways. Further application of this dimerization to hitherto unknown bis-amino(aryl)carbenes yields air-stable, non-dynamic, electron-rich N-PPV's following a "dimerizing polymerization" pathway. Of particular interest and in contrast to classical PPV's, the properties of N-PPV's can readily be manipulated upon protonation, opening the avenue to PPV's with tunable absorption or showing switchable "on-off" absorption.

## Results and discussion

Acyclic MACs, and in particular amino(aryl)carbenes, have been accessed by different synthetic routes, including substitution of the phosphonio group at the carbene center of amino-phosphoniocarbene using aryl-lithium as nucleophiles, reduction of phosphonio aldiminium salts, or dechlorination of C-chloro iminiums with $Hg(TMS)_2$[14,16,45]. Because related methodologies involve either rather sophisticated precursors or toxic reagents, we turned our attention to the deprotonation of aldiminium salts in presence of a strong base[11]. For that purpose, we kept in mind that methyl group at the nitrogen atom should be avoided to suppress any carbene to azomethine ylide isomerization[46]. The substitution pattern of the aryl moiety, especially at the ortho positions, has also been shown to dramatically influence the kinetic stability of the carbene, which eventually controls its deactivation pathways (dimerization versus C–H insertions)[11,46,47].

**In-situ generation of amino(aryl)carbene 2a–c.** On this basis, mono-iminiums **1**, i.e. featuring a di-isopropylamino/phenyl (**1a**), piperidyl/phenyl (**1b**) or piperidyl/fluorenyl (**1c**) substitution pattern, were selected and synthesized according to established procedures (Fig. 2a)[48,49]. LiHMDS was initially investigated for the deprotonation of **1a-c** at −80 °C in THF. Monitoring these reactions by [1]H NMR spectroscopy evidenced that quantitative conversions of **1a-c** into the corresponding diaminoalkenes, **3a-c**, were achieved, highlighting the high selectivity of those transformations. Upon purification, **3a-c** were obtained in good yields (60–70%, see Supporting information for details) as a mixture of E/Z isomers in a molar ratio of 80/20 for **3a** and 85/15 for **3b** and **3c**, respectively. Alternatively, a neutral organic strong base, such as phosphazene $P_4$-tBu, could also be successfully employed instead of LiHMDS, thus providing a transition metal-free synthetic route to **3a-c**.

**Characterization of 3a by X-ray diffraction analysis.** These compounds were thoroughly characterized by [1]H and [13]C NMR spectroscopy, but also by X-Ray diffraction analysis (see Supporting Information for details). In the case of **3a**, single crystals could be grown from a pentane solution of **3a** at −20 °C. Under those conditions, only the E-isomer was found to crystallize (**3a–E**; Fig. 2b). The asymmetric unit of **3a-E** contains two independent molecules. However, only one molecule is described as they both have similar geometrical parameters ($C_3$-$N_1$ and $C_4$-$N_2$ bonds of 1.42 Å and 1.41 Å respectively; C=C bond of 1.359 Å and dihedral angles $C_1$-$C_2$-$C_3$-$C_4$ of 68° and $C_3$-$C_4$-$C_5$-$C_6$ of 62°). X-Ray analysis revealed that the $C_3$=$C_4$ bond distance (1.354 Å) was slightly longer than that of trans-stilbene (1.318 Å), while the $C_3$-$N_1$/$C_4$-$N_2$ bond lengths (1.41 Å and 1.41 Å, respectively) proved shorter than a single C-N bond (1.469 Å). These data thus suggested some interaction between the lone pairs on nitrogen atoms and the central double bond in the solid state[50,51].

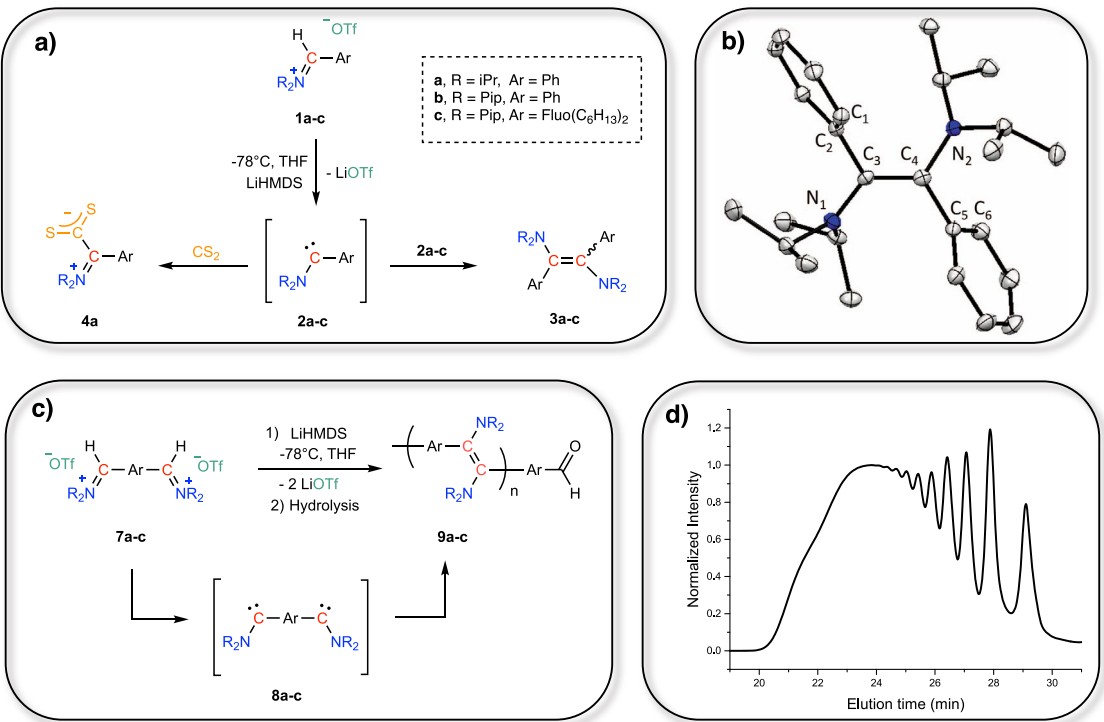

**Fig. 1 Synthetic strategy to amino-poly(phenylene vinylene)s (N-PPV's).** NPPV's via dimerizing polymerization of in situ generated bis-amino(aryl) carbenes, followed by the reversible switching of their physicochemical properties through simple protonation (*B* base).

**Fig. 2 Reactivity of in situ generated mono- and bis-amino(aryl)carbenes. a** Dimerization of mono-carbenes 2a-c and trapping of 2a. **b** Ortep drawing of the molecular structure of dimer **3a-E**. Hydrogen atoms are omitted for clarity. Carbon atoms are represented in gray and nitrogen atoms in blue. Thermal ellipsoids are shown at 50% probability. **c** In situ generation of bis-carbenes 8a-c and formation of the corresponding N-PPV's 9a-c: iPr = isopropyl, Pip = piperidine, Ph = phenyl, Fluo = 9,9-dihexyl-9H-fluorene (see Fig. 1). **d** Size exclusion chromatography trace in THF of N-PPV 9b ($M_n = 3900$ g mol$^{-1}$; $Đ =$ 2.05; calibration with PS standards).

Furthermore, in comparison with the near planarity of trans-stilbene (dihedral angle $C_1$-$C_2$-$C_3$-$C_4$ of 5°), phenyl groups were found significantly tilted in **3a-E** (dihedral angles $C_1$-$C_2$-$C_3$-$C_4$ of 68° and $C_3$-$C_4$-$C_5$-$C_6$ of 69°), to accommodate the steric congestion imposed by the bulky di-isopropylamino groups.

**Characterization of 3 by UV/Vis spectroscopy analysis.** In chloroform solution, this interaction between the amino groups and the central C=C double remained, as evidenced by the strong absorption in the 388–440 nm range observed by UV/Vis analysis of

**3a-c** (see Supporting Information for details), which is red-shifted in comparison with that of trans-stilbene ($\lambda_{max} = 322$ nm)[51]. According to DFT calculations (B3LYP/def2-SVP), this absorption corresponds to the HOMO-LUMO transition, the HOMO being mainly localized both on $\pi_{C=C}$ and $n_N^-$ orbitals and the LUMO being a combination of $\pi^*_{C=C}$ with $\pi^*_{C=C(phenyl)}$ orbitals (see Supplementary Figs. 22, 23 and Supplementary Table 2 for details).

**Trapping of carbene 2a with CS$_2$.** Despite several attempts to characterize **2a** by monitoring the deprotonation of **1a** by $^{13}$C

NMR spectroscopy at −80 °C, no carbene could be identified, as the characteristic low-field signal expected around 300 ppm in the $^{13}$C NMR spectrum could not be detected. Nevertheless, betaine **4a**, resulting from the trapping of carbene **2a** with $CS_2$ (Fig. 2a), was fully characterized by $^1$H, $^{13}$C NMR, and X-Ray diffraction analyses (see Supplementary Information), which strongly supports the in situ generation of **2a**.

**DFT calculation**. Although the dimerization of (cyclic) diaminocarbenes has been theoretically investigated[13,15,17], data concerning MACs, and particularly amino(aryl)carbenes, are lacking. Therefore, to gain some insight into the selective dimerization of carbenes **2a**, relative to putative intramolecular C-H insertion reactions[47], DFT calculations were carried out on the direct dimerization of **2a**, as well as on the intramolecular C-H insertion of **2a** into CHiPr and $CH_3$iPr at the B3LYP/def2-SVP level of theory. The β-H elimination of propene from $i$Pr$_2$N-substituted carbene **2a** was also investigated in the Supplementary Information. Furthermore, computational details, as well as in depth discussion of structural parameters relative to transition states involved in those processes can be found in the Supplementary Information. As anticipated, carbene **2a** was found to possess a singlet ground state that lays 18.5 kcal mol$^{-1}$ below the triplet state. The $C_{carbene}$-N bond proved short (1.304 Å) and the nitrogen atom was found in a planar environment ($\Sigma_N = 360.0°$), indicating a strong interaction between the nitrogen lone pair and the carbene empty orbital. Furthermore, the acute $NCC_{Ph}$ angle (124.2°) and the long $C-C_{Ph}$ bond distance of 1.457 Å attested that the phenyl group merely behaved as a mesomeric spectator.

**Investigation of the dimerization of 2a**. Dimerization reactions of **2a** into the corresponding diaminoalkenes **3a-E** and **3a-Z** were next computed, as summarized in Fig. 3 (see also the Supplementary Fig. 24 and Supplementary Table 5). From a kinetic point of view, moderate activation barriers (18.5 and 17.2 kcal mol$^{-1}$ for **2a**→**3a-E** and **2a**→**3a-Z**, respectively) were calculated. In addition, both transformations were predicted strongly exergonic ($\Delta G = -45.0$ and $-35.3$ kcal mol$^{-1}$ for **2a**→**3a-E** and **2a**→**3a-Z**, respectively), the relative stability of **3a-E** vs. **3a-Z** being in agreement with experimental observations (**3a-E** and **3a-Z** being the thermodynamic and kinetic product, respectively).

**Investigation of putative insertion reactions of 2a**. In the case of the putative transformations of **2a** into cyclopropane **5a** and cyclobutane **6a**, via carbene insertion into CHiPr and $CH_3$iPr, respectively, both reactions were predicted to be highly exergonic (−16.0 and −30.6 kcal mol$^{-1}$, respectively). However, activation barriers of 39.4 kcal mol$^{-1}$ and 45.6 kcal mol$^{-1}$ were calculated for the insertion of **2a** into CHiPr and into $CH_3$iPr, respectively (see Fig. 3 and the Supplementary Fig. 25 and Supplementary Table 6). In other words, both insertion reactions proved prohibitively too high to compete with the dimerization process under our experimental conditions.

**Dimerizing polymerization of in-situ generated bis-carbenes 8a-c**. The high conversion and complete selectivity in favor of the formal dimerization of **2a-c** into the corresponding diaminoalkenes **3a-c** prompted us to design original bis-aldiminium salts **7a-c** and to investigate their deprotonation (see Fig. 2c). Indeed, the resulting in situ generated bis-carbenes **8a-c** would thus serve as difunctional monomeric substrates and would lead to an original class of PPV's, namely amino-PPV's (N-PPV's), which can be hardly accessed by other means, here following a dimerizing step-growth polymerization pathway. Bis-aldiminiums **7a-c** were thus prepared by adapting the procedures followed for the preparation of mono-iminiums **1a-c**.

**Characterization of N-PPV's 9a-c**. Bis-aldiminiums **7a-c** displayed similar NMR spectra, with very few signals due to their symmetrical nature. For instance, while **7a** possesses 34 protons, only 6 different signals were observed in its $^1$H NMR spectrum. Most notably, the characteristic $CH_{iminium}$ appeared as the most downfield signal at 9.5 ppm (172 ppm in $^{13}$C NMR) and only a singlet at 7.86 ppm, integrating for 4H, was observed for the *para*-disubstituted phenyl group. Upon deprotonation of **7a-c** using either LiHMDS or P$_4$-$t$Bu in THF, a gradual red color appeared, suggesting the formation of the targeted conjugated N-PPV's **9a-c** (Fig. 2c). Remarkably, those polymers proved air-stable and do not exhibit any dynamic behavior, in contrast to conjugated polymers derived from bis-benzimidazolylidenes[29–31]. This difference also highlights how the substitution pattern of the carbene center dictates, not only the structure but also the properties of the resulting polymers. SEC analysis revealed typical traces of a step-growth polymerization process (see Fig. 2d for a representative SEC trace), with multimodal distributions composed of discrete peaks corresponding to

**Fig. 3 Theoretical investigation of the reactivity of 2a.** General view of the energetic profiles for the dimerization of carbene 2a into diaminoalkenes 3a-E/ Z and for the putative intramolecular CHiPr and $CH_3$iPr insertion reactions (see Supporting Information for details), at the B3LYP/def2-SVP level of theory; Gibbs free energy are expressed in kcal. mol$^{-1}$.

oligomers: $M_n = 3000$ (dispersity, $Đ = 2.08$), 3900 ($Đ = 2.05$), and 5200 g mol$^{-1}$ ($Đ = 2.60$) for **9a**, **9b**, and **9c**, respectively (Supplementary Figs. 4–6). Importantly, molar masses of **9a** and **9c** determined by MALDI ToF mass spectrometry were found in rather good agreement with values delivered by SEC, with DP = 5-6 and DP = 10 for **9a** and **9c**, respectively (see Supplementary Figs. 20 and 21). Furthermore, 3 distinct populations could be identified, in the case of **9c**, with different chain ends. Among these, one population corresponded to the expected bis-aldehyde-terminated N-PFV, owing to the hydrolysis of both iminium/carbene chain-ends chain during workup, while another population was ascribed to a N-PFV carrying an iminium and an aldehyde chain end. Interestingly, cyclic N-PFV structures consisting of 3–9 repeating units were also identified in the MALDI ToF mass spectrum of **9c** (see Supplementary Fig. 20).

In comparison with the $^1$H NMR spectra of diaminoalkenes **3a-c**, N-PPV's **9a-c** displayed signals with similar chemical shifts, but showed evident broadening due to their polymeric nature. The only sharp signal observed at 10.0 ppm in the $^1$H NMR spectrum of **9a-c** corresponded to aldehyde end-groups, arising from the hydrolysis of carbene/iminium chain ends during workup in air, in agreement with structures determined by mass spectrometry. Importantly, similar E/Z ratios of 80-85/ 20-15 were determined for both **9** and **3**, suggesting independent reactivity of each carbene site in the bis-carbene monomers. It should be noted that higher molecular weight N-PPV's could not be reached despite the investigation of several experimental conditions, including variations in the nature of both the solvent and the base, and in the concentration (see Supplementary Table 1). Those limitations may stem from the building of steric hindrance around the carbene center upon repeated dimerization reactions. In parallel, DFT calculation suggested that β-H elimination of propene from iPr$_2$N-substituted carbene **2a** could compete with the dimerization reaction ($\Delta G = -18.5$ kcal mol$^{-1}$ and $\Delta G^{\#} = 19.1$ kcal mol$^{-1}$; Supplementary Figure 26 and Supplementary Table 7). In the case of piperidine-substituted carbenes, such a deactivation pathway should not be operative owing to the geometrical constraint imposed by the cyclic structure of the stabilizing amino group.

Finally, the increased conjugation in polymers **9a-c**, respectively to diaminoalkenes **3a-c**, could also be appreciated from the red-shift (43–76 nm) of the maximum absorption by UV/Vis analysis (Fig. 4). By analogy with molecular diaminoalkenes **3a-c**, the interaction between the amino groups and the C=C bond in N-PPV's **9a-c** resulted in a bathochromic effect on the $\lambda_{max}$ compared to the parent PPV (440–483 nm vs. 400–420 nm for

N-PPV's and PPV's respectively)[52,53]. Furthermore, the large stoke shifts observed (see Supplementary Figs. 17–19) are promising considering their possible integration into light-emitting devices.

**Film-forming properties of N-PPV's.** The film-forming properties of N-PPV's **9** were also briefly investigated. Gratifyingly, while the formation of PPV's film generally requires high molar mass, a homogeneous film with a thickness of 85 nm could be prepared by spin-coating a 15 mg/mL chloroform solution of **9c** onto a glass substrate (Supplementary Figs. 1–4). The film stability was evidenced by AFM and optical microscopy.

**Tuning the absorption of N-PPV's 9 by protonation.** While tuning the properties of classical PPV's is challenging, the presence of amino groups in α-position of the C=C bonds in N-PPV's **9a-c** provides facile means to vary the properties of this organic polymeric platform, upon chemical post-modification (Fig. 5).

Thus, in a preliminary experiment, treatment of **9b** with 2 eq. of TfOH at −40 °C in a CH$_3$CN/CHCl$_3$ mixture (50/50) led to the disappearance of the maximum absorption at 476 nm, suggesting that the conjugation was disturbed in **10b** (see Fig. 5a). Interestingly, this process was found to be fully reversible and the maximum absorption could be recovered in **9'b** upon deprotonation of **10b** using *t*-BuOK at 0 °C. The identity of **9'b** was also confirmed by $^1$H NMR analysis. While a similar behavior was observed with **9c**, a completely different scenario was evidenced with **9a**. In the latter case, the maximum absorption initially at 462 nm was blue-shifted to 352 nm upon protonation (Fig. 5b). This different behavior may be surprising as **9a** and **9b** only differ by the nature of the amino substituents on the C=C bond (di-isopropylamino *vs*. piperidine in **9a** and **9b**, respectively). To gain some insight into the different behavior of **9a** and **9b**, protonation of the corresponding molecular dimers, **3a** and **3b**, was then carried out using 2 eq. of TfOH (see Supplementary Information). According to X-ray diffraction analysis, protonation of **3a** occurred at both nitrogen atoms, the C=C bond being preserved. In sharp contrast, **3b** underwent protonation at only one nitrogen atom and at the enamine carbon atom, ultimately leading to a dicationic compound without C=C bond (see Supplementary Information for details). By analogy, we thus hypothesized that protonation of N-PPV's **9a** and **9b** would lead to conjugated poly(ammonium) derivative **10a** (Fig. 5b) and non-conjugated **10b** (Fig. 5a), respectively, in agreement with a blue-shift and the disappearance of the $\lambda_{max}$ in the UV/vis spectrum of **9a** and **9b** upon protonation.

Overall, this study demonstrates that weakly kinetically stabilized acyclic amino(aryl)carbenes undergo a facile and highly selective dimerization reaction, yielding the corresponding diaminoalkenes. Of particular interest, in situ generated bis-amino(aryl)carbenes lead to the formation of air-stable N-PPV's by a dimerizing step-growth polymerization route. Importantly, N-PPV's reported herein cannot be prepared by traditional routes. Furthermore, and in contrast to more conventional PPV's, the presence of amino groups adjacent to the C=C double bonds in N-PPV's allows their properties to be manipulated in different ways upon protonation, depending on the nature of the substituent on the nitrogen atoms. Thus, this work could pave the way toward PPV's featuring tunable absorption and/or to sensor-type materials thanks to the ability of N-PPV's to readily form stable films. Future work is also underway to expand the synthetic interest of this under-explored class of mono-amino carbenes in the context of polymer chemistry, through a careful and more systematic design of the substitution pattern of the carbene center.

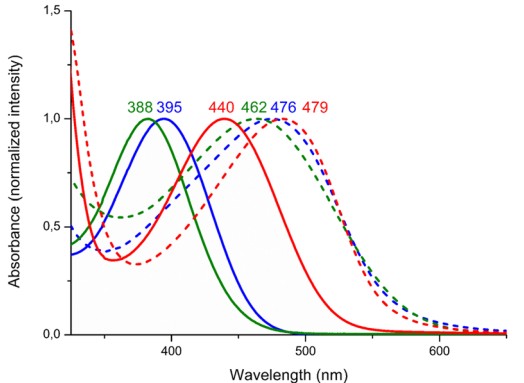

**Fig. 4 UV/Vis spectrometry of dimers 3 and corresponding N-PPV's 9 in CHCl$_3$.** Solid lines, green **3a**, blue **3b**, red **3c**; dashed lines, green **9a**, blue **9b**, red **9c**.

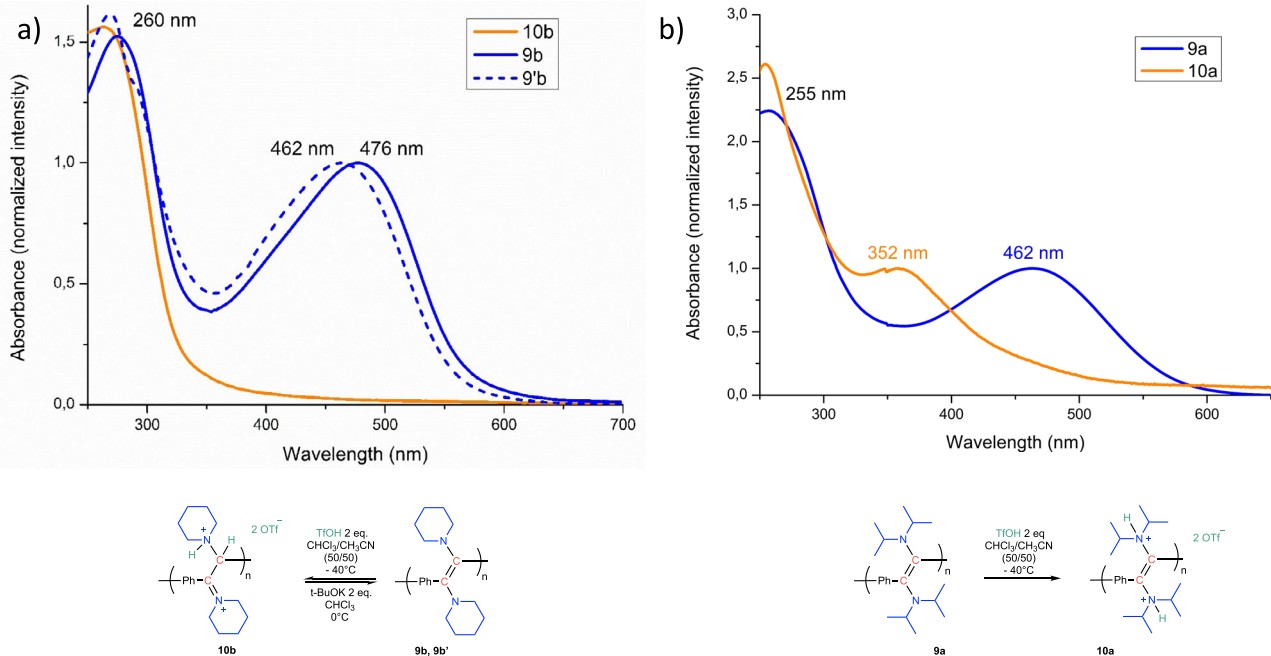

**Fig. 5 Switching of the absorption of N-PPV's 9. a** Protonation of N-PPV **9b** (solid blue line) using TfOH and subsequent deprotonation of polymer **10b** (solid orange line) in presence of t-BuOK leading to N-PPV **9'b** (dashed blue line), along with their respective UV spectra (**9b**, **9'b** and **10b**). **b** Protonation of N-PPV **9a** using TfOH and formation of poly(cation) **10a** and their corresponding UV spectra (**9a** and **10a**).

## Methods

**Materials**. THF and $Et_2O$ were dried over sodium/benzophenone and distilled prior to use. Dichloromethane, chloroform and pentane were dried over $CaH_2$ and distilled prior to use. Acetonitrile solvent was dried using a MBraun Solvent Purification System (model MB-SPS 800) equipped with alumina drying columns. Benzaldehyde (Alfa Aesar, 98%) was dried over $CaH_2$, distilled and stored under argon. n-Butyllithium solution 11 M in hexanes, Lithium bis(trimethylsilyl)amide, Phosphazene ($P_4$-tBu) 0.8 M in hexane and terephthalaldehyde were purchased from Sigma-Aldrich and used without further purification. Diisopropylformamide (Alfa Aesar) and piperidinoformamide (Sigma Aldrich) were stored under argon over activated molecular sieves 4 Å. Trifluoromethanesulfonic acid (TCI), Trifluoromethanesulfonic anhydride (ABCR), and benzaldehyde (Alfa Aesar) were distilled prior to use. Phenyllithium 1.9 M in di-n-butyl ether was purchased from Alfa Aesar and used without purification.

**Instrumentation**. NMR spectra were recorded on a Bruker Avance 400 (1H, 13 C, 19 F, 400.2, 100.6, and 376.53 MHz, respectively) in appropriate deuterated solvents. Molar masses were determined by size exclusion chromatography (SEC) in THF + $Et_3N$[54–56] (1 mL/min) with trichlorobenzene as flow marker, using both refractometric (RI) and UV detectors. Analyses were performed using a three-column TSK gel TOSOH (G4000, G3000, G2000) calibrated with polystyrene standards. HRMS were recorded on various spectrometer: a Waters Q-TOF 2 spectrometer and a GCT premier water mass spectrometer in the chemical ionization mode. Melting point were determined by using a Stuart Scientific SMP3 apparatus. Ultraviolet–visible spectra were collected on a thermostated UV/Vis Spectrometer (Agilent Carry 4000). Fluorescence spectra were obtained via Spectrofluorimeter (Jasco FP-8500ST). XRay structures were done on a Rotating Anode Rigaku FRX 3 kW with microfocus (Hybrid Dectris Pilatus 200 K pixel detector).

**Synthesis of aldiminium salt precursors**. A typical procedure to synthesize benzylidenediisopropyliminium triflate, **1a**, is as follows. *Following the Alder's route*: To a stirred solution of diisopropylformamide (1.38 mL, 9.5 mmol) in dry ether (50 mL) cooled at −78 °C was added phenyllithium (1.9 M, 7.9 mmol) dropwise and the resulting mixture was stirred at this temperature for 30 min then at room temperature for 1 h. Then, to the reaction mixture cooled at −78 °C was added trifluoromethanesulfonic anhydride and the mixture was stirred for 1 h at this temperature then for 2 h at room temperature. The precipitated solid was filtered under argon and washed with dry ether (3 × 5 mL) and THF (3 × 5 mL) and dried under vacuum to obtain compound **1a**, as white crystals (2.1 g, 70%). *Following the Schroth's route*: To a stirred solution of benzaldehyde (1 mL, 10 mmol) and 1-(trimethylsilyl) diisopropylamine (1.45 g, 10 mmol) in dry ether (50 mL), TMSOTf (1.8 mL, 10 mmol) was added dropwise at room temperature and the resulting mixture was stirred at the same temperature for 6 h under inert atmosphere. The precipitated solid was filtered under inert atmosphere and dried under vacuum to

obtain compound **1a** as white crystals (2.32 g, 70%). See the Supplementary Information for a complete characterization.

**Dimerization reaction from monofunctional adiminium salts**. A typical procedure leading to the diamino-containing alkenes, **3a**, is as follows. *Using $P_4$-tBu base*: To a stirred solution of **1a** (0.1 g, 0.3 mmol) in THF (6 mL) cooled at −78 °C, $P_4$-tBu base (0.38 mL, 0.3 mmol, 0.8 M in hexanes) was added and the resulting mixture was stirred at this temperature for 1 h followed by 24 h at room temperature. The solvent was evaporated under reduced pressure and the reaction mixture was diluted with pentane (20 mL) and passed through a short column of basic alumina. The final compound **3a** (0.034 g, 60%) was obtained as a yellow solid. Single crystals were grown from the slow evaporation of a pentane solution of **3a** at −20 °C. *Using LiHMDS base*: **1a** (0.1 g, 0.3 mmol) and LiHMDS (50 mg, 0.3 mmol) were dissolved in cold THF (6 mL, −78 °C). The resulting mixture was stirred at this temperature for 1 h followed by 24 h at room temperature. The solvent was evaporated under reduced pressure and the reaction mixture was diluted with pentane (20 mL) and passed through a short column of basic alumina. The final compound **3a** (0.034 g, 60%) was obtained as a yellow solid. Single crystals were grown from the slow evaporation of a pentane solution of **3a** at −20 °C. See the Supplementary Information for a complete characterization.

**Synthesis of amino-containing poly(phenylene vinylene)s (N-PPV's) by dimerizing polymerization from in situ generated bis-aminoaryl carbenes**. A typical procedure for achieving N-PVV **9a** is as follows. To a stirred solution of **7a** (0.300 g, 0.5 mmol) in THF (2 mL) cooled at −78 °C $P_4$-tBu base (1.38 mL, 1.1 mmol, 0.8 M in hexanes) was added and the resulting mixture was stirred at this temperature for 1 h followed by 24 h at room temperature. After the solvent was removed under reduced pressure, the mixture was washed with MeOH (10 mL) to remove the phosphazenium salts. The resulting mixture was dissolved in $CHCl_3$ (15 mL) and filtered to remove the insoluble compounds. After evaporation of the organic solvent, the mixture was purified by precipitation using $CHCl_3$ and MeOH at −10 °C to obtain the desired polymer **9a** as a dark red solid (80 mg, 60%). See the Supplementary Information for a complete characterization and for an optimization study.

**Post-chemical modification of N-PPV's**. A typical protonation of N-PPV **9** is as follows. To a solution of polymer precursor **9b** (0.134 g, 0.50 mmol) in $CH_3CN$/$CHCl_3$ (50/50) cooled at −40 °C, TfOH (184.2 µL, 2.2 mmol) was added dropwise. The resulting mixture was warm to room temperature overnight and the organic solvent was evaporated to give the compound **10b**, namely, poly(7,8-bispiperidinium-1,4-phenylenevinylene) bis-triflate: (99%, 0.245 g); UV/Vis ($CH_3CN$) $\lambda_{max}$ = 254 nm.

Deprotonation of polymer **10b** is as follows. To a stirred solution of **10b** (0.054 g, 0.1 mmol) in $CH_3CN$/$CHCl_3$ (50/50) cooled at 0 °C, t-BuOK (0.021 g, 0.2 mmol)

was added. The reaction was warmed to room temperature overnight, and the solvents were removed under reduced pressure and the resulting mixture was extracted with $CHCl_3$ leading to the final polymer **9b** (0.048 g, 89%).

## Data availability

The X-ray crystallographic coordinates for structures reported in this study have been deposited at the Cambridge Crystallographic Data Centre (CCDC), under deposition numbers 1973046 for **7a** 1973048 for **3a**, 1973057 for **3b**, 1973059 for **3c**, 1973049 for **4a**, 1973058 for **E-3b(HOTf)₂** and 1973060 for **E-3a(HOTf)₂**. These data can be obtained free of charge from The Cambridge Crystallographic Data Centre via www.ccdc.cam.ac. uk/data_request/cif. The authors declare that all other data supporting the findings of this study, including computational details, are available within the article and Supplementary Information files, and also are available from the corresponding author on reasonable request.

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

## Acknowledgements

The authors are grateful to CNRS and to the Agence Nationale de la Recherche (ANR): CARBENOPOL Project (ANR-19-CE06-0015) for financial support. This research was also supported by LabEx AMADEus (ANR-10-LABEX-0042-AMADEUS). We thank Ms. A.–L. Wirotius (LCPO) for NMR analysis and Dr. B. Kauffmann (IECB) for X-Ray diffraction analysis. G. Pino and G. Pécastaing are warmly acknowledged for their help in evidencing the formation of stable films using AFM and optical microscopies. UPPA, MCIA (Mesocentre de Calcul Intensif Aquitain), and CINES under allocation A009080045 made by "Grand Equipement National de Calcul Intensif" (GENCI) are acknowledged for computational facilities. CDAPP is also thanks for funding part of the post-doctoral contract of E. Daiann Sosa Carrizo.

## Author contributions

J.V. and D.T. designed and supervised the investigations. Q.S., A.K., and D.M. performed the synthesis of (macro)molecular compounds. D.S.C., K.M., and J.-M.S. performed DFT calculations. E.C. and C.B. were in charge of the characterization by absorption/emission. Q.S., A.K., D.M., D.S.C., K.M., J.-M.S., E.C., C.B., Y.L., D.T., and J.V. were involved in the analysis of data and their interpretation. J.V. and D.T. wrote the manuscript with the input of all other authors. All the authors agreed on the final manuscript.

## Competing interests

The authors declare no competing interests.
