## [Peer Review File · Nature Communications]

REVIEWER COMMENTS

Reviewer #1 (Remarks to the Author):

I reviewed a previous version of this manuscript for a different journal. The manuscript is largely unchanged and so my comments and opinion still stand. While it does appear that the authors have addressed some of the issues that were raised, others are outstanding. As such, notes have been peppered into the passage to provide further clarification and context. Overall, this is an excellent paper but, in its current form, it is oversold and not placed into proper context. Additional work is needed before it can be published, in my opinion.

--

PREVIOUS REVIEW WITH NOTES

This manuscript describes the synthesis of amino substituted poly(phenylene vinylene)s (N-PPVs) using carbene dimerization methodology. A series of acyclic aminoaryl carbenes were generated in situ which resulted in spontaneous dimerization to afford the corresponding diaminoethylenes. The polymerization reactions were guided by model studies of monotonic (thus, non-polymerizable) variants as well as comprehensive calculations. The N-PPVs were found to exhibit relatively bathochromic absorbances when compared to the model compounds, and could be reversibly modulated through chemical treatment. As such, the polymeric materials may hold potential value for use in optoelectronic applications.

Overall, this is a nice paper, but a bit oversold and apparently unaware of literature precedent.

For example, the statement "The development of singlet carbenes ... as true building blocks for polymer synthesis has been overlooked" is categorically untrue. In fact, reviews on the topic are now available. To inform the authors and future readers, the following papers should be added to the references: (a) JACS 1997, 6668; (b) JACS 2005, 12496; (c) Organometallics 2006, 6087; (d) Chem. Soc. Rev. 2007, 729; (e) Macromolecules 2010, 43, 6923; (f) etc. [NOTE: perplexingly, the authors have added the references but did not correct the misleading statement.]

As another example, the statement "...the presence of amino groups in the ... N-PPV's provides an innovative mean to vary the properties ... upon chemical post-modification" is also precedented; see: Chem. Commun. 2009, 2124. [NOTE: this issue remains unaddressed.]

Other points of concern pertain to the N-PPVs.

The polymers were of relatively low molecular weight which, as the authors note, can be attributed to inherent limitations of step-growth based methodologies. Regardless, the potential of conjugated polymers rests in their ability to form films. Can films of the N-PPVs reported be prepared? The polymerization methodology may need to be optimized to achieve sufficiently high molecular weights. [NOTE: the conclusion now states that future efforts will entail the synthesis of higher MW polymers and potentially films; however, it seems imperative to me that the feasibility of such aims need to be realized in order for the methodology to attract wide appeal.]

In addition, while the N-PPVs do appear to absorb lower energy light (~475 nm) than their analogous dimers (~400 nm) which may reflect an increased conjugation length, do the electronic signatures improve upon known PPVs? For example, MEH-PPV, which is commercially available, absorbs at a similar wavelength at the N-PPVs and can be prepared in molecular weights that exceed 50 kDa and thus readily form films. MEH-PPV and other electron rich PPVs are also photoemissive and afford high quantum yields. Additional data and/or clarification are needed. [NOTE: this issue remains unaddressed. The authors are encouraged to provide clear statements that tout the potential advantages of the N-PPVs versus known PPVs (e.g., MEH-PPV, etc.) to inform readers of the significance of the work described.]

Overall, this manuscript is off to a good start and perhaps with further modification, a well rounded paper that is broadly appealing may emerge.

Other issues include:

The passage "...reported that peculiar bis-benzimidazol-2-ylidene (sic) can undergo..." should be clarified as ref. 11 reports multiple bis(NHC)s which undergo dimerization. [NOTE: this issue appears to have been addressed.]

The statement "...are also air-sensitive, which makes them not best suited for applications" is misinformed and misleading. Many conjugated polymers (e.g., polyacetylene, polypyrrole, etc.) are air sensitive yet have found utility in a broad range of applications. [NOTE: this issue appears to have been addressed.]

Reviewer #2 (Remarks to the Author):

The manuscript reports a polymerization through the dimerization of singlet carbene species. The concept is interesting, but I believe the work has not reached the point for publication. In particular, for the very limited scope of polymerization the molecular weights, 3000 g mol⁻¹, 3900 g mol⁻¹ and 5200 g mol⁻¹ for 9a, 9b and 9c, are pretty low and there is no information about polymerization optimization. The molecular weights may be much improved by increasing the concentration of monomers or reaction time, etc. And what is the main side reaction of this polymerization resulting in the low molecular weight? Heteroaryl on the main chain plays an important role in the conjugated polymers with optoelectronic properties. I am interested in whether any experiment is done on the acyclic amino(heteroaryl)carbene and its polymerization. The scope of the substances may need to be further broaden. The presence of amino groups in alpha-position of C=C double bonds in N-PPV's allows their physico-chemical properties to be reversibly manipulated upon chemical post-modification. Whether this phenomenon is just suitable for 9b, or chemical post-modification of the other two polymers, 9a and 9c, leads to the same results. And I am interested in how color of the solution changed during protonation/deprotonation.

Other points.

(1) The size exclusion chromatography spectra described in Fig. 2 (D) with poor resolution, which can not attain publishable standard. Please revise this point. Figure S10 is with the same problem.

(2) There are some mistakes in Fig. 4.

a. The color of the chemical structure of a, b, c dose not match the line color of 3a, 3b, 3c and 9a, 9b, 9c.

b. Vertical coordinate of Fig. 4 (B) is not in accordance with that of Fig. 4 (A).

c. The end of UV spectra of polymers should infinitely approach zero like that of Fig. 4 (B).

d. In the chemical equation of Fig. 4 (B), the reaction conditions are not in agreement with the description in page 6. Please check it again.

(3) Some attention needs to be paid to the References Section. Upon looking up the references, I noticed format of some references are wrong.

a. The hyphens, like in Re 8, 23, 25 etc, should be corrected.

b. In Ref 33, 'Polym. Rev ' should be 'Polym. Rev. '.

It is possible other errors in the References Section.

(4) In the Materials and Methods Section, format of describing reactions should be paid much more attention, such as '3x5mL ', '0.048g ', '78°C '.

(5) Emission spectrum of compounds 9b (Figure S8) in thin film is obviously unnormal, the end of the spectrum is almost vertical to the coordinate. It is necessary to collect this emission spectrum again or demonstrate this phenomenon. What' more, why are the data of emission spectra of polymers in solution lacking?

Reviewer #3 (Remarks to the Author):

Manuscript NCOMMS-20-18745 discloses the preparation of some monoaluminum salts (Ar-C(H)-

NR₂(OTf) (1a-c) which were deprotonated using strong bases to yield the acyclic (amino)(aryl)carbenes. These carbenes dimerized to give the diaminoalkenes 3a-c. Next aryl-bridged bis-aldiminium salts 7a-c were prepared. Deprotonation of these derivatives yielded bis-carbenes which upon carben-carben interaction yielded polymers (amino-PPVs). While one of the dimeric bis-carbene compounds (E-3a) was characterized by an X-ray diffractions study revealing some differences relative to the known E-stilbene structure. Characterization of the polymers prove more difficult but is convincingly presented and supported by NMR spectroscopy.

My major concerns with this manuscript is the stability of the obtained dimers and polymers. Dimerization of two stable singlet carbenes could lead to trans-bent double bonds (refs 19, 21, 22) and these are known not to be very stable (see ref 22 for an equilibrium between NHC and its enteramine). If the authors can satisfactorily address the question of stability of their dimers and polymers, I believe that the manuscript meets the standards of Nat. Commun. In addition, some typos (bold face for compounds in Figure 3a and in other graphics) should be corrected. Finally, ref 47 should be changed to state correctly that the asymmetric unit of 3a-E contains two independent molecules.

Direct and Selective Access to Amino-Poly(Phenylene vinylenes)s with Switchable Properties by Dimerizing Polymerization of Aminoaryl Carbenes

Manuscript NCOMMS-20-32100: Answers to reviewers' comments

Reviewer#1

General comment. "I reviewed a previous version of this manuscript for a different journal. The manuscript is largely unchanged and so my comments and opinion still stand. While it does appear that the authors have addressed some of the issues that were raised, others are outstanding. As such, notes have been peppered into the passage to provide further clarification and context. Overall, this is an excellent paper but, in its current form, it is oversold and not placed into proper context. Additional work is needed before it can be published, in my opinion."

Answer. We thank the reviewer for the rather positive tone of these comments. As mentioned in the cover letter above, and as discussed in more details in the point-to-point answers below, novel experiments have been performed and the manuscript has been revised accordingly.

Comment. "For example, the statement "The development of singlet carbenes ... as true building blocks for polymer synthesis has been overlooked" is categorically untrue. In fact, reviews on the topic are now available. To inform the authors and future readers, the following papers should be added to the references: (a) JACS 1997, 6668; (b) JACS 2005, 12496; (c) Organometallics 2006, 6087; (d) Chem. Soc. Rev. 2007, 729; (e) Macromolecules 2010, 43, 6923; (f) etc. [NOTE: perplexingly, the authors have added the references but did not correct the misleading statement.]"

Answer. In fact, we had already changed this statement from the paper that was initially submitted in another journal -and that was evaluated by the same reviewer- to the version submitted to *Nature Communications*. In the former case, we wrote: "The development of singlet carbenes as organocatalysts has led to spectacular achievements both in organic and polymer synthesis. Yet, their utility as building blocks for polymer synthesis has been overlooked." In the version submitted to *Nature Communications*, the Abstract now mentions: "Despite the ubiquity of singlet carbenes in chemistry, their utility as true monomeric building blocks for the synthesis of functional organic polymers has been overlooked."

In the introductory part, and to take into account the reviewer's comment about the citation of the literature background, we have added the following sentence. "Formal polymerization by dimerization of non-heteroatom-stabilized bis-(aryl)carbenes, generated by decomposition of corresponding bis-diazo compounds, has also been reported." We, however, understand the general reviewer's concern about the literature precedent. In fact, references she/he has suggested all deal with the synthesis of organometallic polymers, for which bis-*N*-heterocyclic carbenes (NHCs) have served as ligands of transition metals (M), *i.e.* with NHC-M bonds. Thus, only works by Bielawski *et al.* on the dimerization of bis-NHCs to afford non-PPV conjugated polymers, and works reporting the decomposition of bis-diazo precursors (= ref. 33-36) that generate "classical" PPV's, *i.e.* that do not possess any hetero-atom connected to the C=C bond, utilize bis-carbenes as monomers. One can therefore truly consider that the use of "singlet carbenes as true monomeric building blocks for the synthesis of functional organic polymers has been overlooked". In the course of the revision of our paper, works by Ihara *et al.* (ACS Omega 2020, 4787) have reported the synthesis of PPV's featuring alkoxy carbonyl groups in α -position of the C=C bonds, by a ruthenium catalyzed decomposition pathway, again employing bis-diazo monomeric precursors; this work has been also added in the reference section (see ref 36).

Comment. "As another example, the statement "...the presence of amino groups in the ... *N*-PPV's provides an innovative mean to vary the properties ... upon chemical post-modification" is also precedented; see: *Chem. Commun.* 2009, 2124. [NOTE: this issue remains unaddressed.]"

Answer. We understand the point raised by the reviewer. We acknowledge that Bielawski *et al.* did take advantage of the presence of amino groups in the main chain of their conjugated polymers obtained from bis-NHC precursors. We therefore remove the word "innovative" in our sentence. We wish to point out again, however, that works by Bielawski *et al.* did not describe the synthesis of PPV's but of poly(tetra-aminoalkene)s from bis-benzimidazolydenes as monomers. As such, those polymers contain weakly basic amino groups, as they are linked to an aromatic and a C=C bond within a constrained cyclic structure. Hence, their reactivity towards Brønsted/Lewis acids is expected to be disfavored relative to that of *N*-PPV's we report here. As a matter of fact, the authors have evidenced that poly(tetra-aminoalkene)s can be oxidized by I₂ at the C=C bonds, affording the corresponding non-conjugated polycationic structure. In contrast, *N*-PPV's reported in our work feature two bulky basic di-alkylamino groups connected to the C=C bonds. Depending on the nature of the amino substituents (piperidine vs. di-*isopropyl*amino), two distinct behaviors were noted. Protonation of **9b**, *i.e.* with piperidine substituents on the C=C bond, with 2 eq. of TfOH led to complete breaking of the conjugation, as observed by UV/visible spectroscopy with the disappearance of the $\lambda_{\text{max}} = 476$ nm. A novel series of experiments performed with the *iso*-propyl derivative, denoted as *iPr*₂N-PPV **9a**, evidenced a completely different behavior. In the latter case, indeed, the λ_{max} at 446 nm was blue-shifted down to 352 nm, highlighting that the conjugation could be preserved to some extent. As detailed below (see response to reviewer# 2), the origin of the difference between **9a** and **9b** likely stems from the bulkiness of the amino groups (piperidine vs. *iso*-propyl). Hence, while **9a** could be protonated at both nitrogen atoms, protonation of **9b** only occurred at one nitrogen atom and, in the meantime, at the enamine carbon atom, leading to a breaking of the conjugation and a complete loss of absorption. It is worthy to note that protonation at the C-site required the N-atom to sp² hybridize (from sp³), which was likely too costly in energy in **9a**, because of the steric hindrance brought by the *iPr*₂N substituent. Overall, these new results demonstrate the interest in positioning basic amino groups adjacent to the C=C bond in our *N*-PPV structures, as their chemical post-modification -here by protonation- shows a dramatic impact on the properties of the pristine polymer, which depends on the steric hindrance around the amino substituents.

These data are presented in the revised manuscript and Figure 5 has been modified accordingly. We hope that these complementary experiments will convince the reviewer about the interest of having two basic amino groups connected to the C=C bonds of *N*-PPV's.

Comment. *The polymers were of relatively low molecular weight which, as the authors note, can be attributed to inherent limitations of step-growth based methodologies. Regardless, the potential of conjugated polymers rests in their ability to form films. Can films of the N-PPVs reported be prepared? The polymerization methodology may need to be optimized to achieve sufficiently high molecular weights. [NOTE: the conclusion now states that future efforts will entail the synthesis of higher MW polymers and potentially films; however, it seems imperative to me that the feasibility of such aims need to be realized in order for the methodology to attract wide appeal.]*

Answer. The reviewer is correct. On this point also, new series of experiments have been carried out in order to achieve higher molecular weights. In particular, the solvent effect has been investigated, as well as the effect of the base (use of an organic vs. a metallic base) mainly from the bis-iminium **7a** (see Table S1 in SI). It is worth emphasizing that the choice of the reaction solvent appears to be limited by the opposite solubility between the dicationic bis-iminium monomeric precursors, and the resulting neutral (non-ionic) *N*-PPV's. Moreover, performing these reactions at concentration higher than 0.1 M (as initially used in our work) led to a heterogeneous mixture, from the beginning to the end of the polymerization, giving very poor yields of the isolated material. In addition to THF that was initially employed, both DMF and toluene were thus studied (entries 2 and 3). However, while DMF proved a good solvent for the bis-iminium precursors, the resulting *N*-PPV's were found to be

hardly soluble in that solvent. The opposite trend was observed upon using toluene. Thus, at the end, neither DMF nor toluene enabled us to reach high molecular weight *N*-PPV's.

Besides this, and by continuing our investigations, we found that a rather weak organic base of phosphazene-type, namely, P₁-^tBu-Pyrr₃, could deprotonate the bis-iminium precursors, but it was not too basic to allow DCM to be used as a polymerization solvent. It is worth reminding that in our initial works, the use of strong bases, such as LiHMDS or P₄-^tBu, was limited to solvents with non-labile hydrogen. Of further interest, the better solubility in MeOH of the conjugated acid H⁺-P₁-^tBu-Pyrr₃ allows for an easier Soxhlet purification of *N*-PPV's. However, deprotonation of **7a** in DCM (0.1 M) in presence of P₁-^tBu-Pyrr₃ afforded a *N*-PPV **9a** showing similar features, *i.e.* of similar molecular weight to those obtained with the other bases (entry 5).

Interestingly, diluting the solution to 0.01 M allowed us to perform these polymerizations under homogeneous conditions, both in DCM and THF as solvents. However, low molecular weight *N*-PPV's (DP = 7-10 for reaction in THF and DCM at 0.01 M) were again achieved, similarly to polymerization performed at 0.1 M in presence of P₁-^tBu-Pyrr₃ (entries 4-5 for DCM and entries 6-7 for THF). Also, diluted solutions enabled to appreciably improve the isolated yields (70-99%).

Lastly, deprotonation of **7a** in THF was attempted by a slow addition process of LiHMDS at a concentration of 0.01 M, with the aim at keeping the number of active carbene sites as low as possible (see entry 10). However, a similar DP of about 10 was again obtained.

Those overall observations suggest that indeed, only *N*-PPV's of limited molecular weights can be obtained, irrespective of the initial experimental conditions.

We also found that SEC analysis in THF of *N*-PPV's could be significantly improved by adding 10% of Et₃N in the eluent, thus affording much better resolved SEC traces (e.g. M_n = 1400-2000 g.mol⁻¹ for iPr₂N-PPV, **9a**). The novel polymerization procedures, as well as a new table reorganizing the different conditions investigated, have been added to the SI (Table S1).

Finally, and to address another important point raised by the reviewer, the ability of *N*-PPV's **9a** and **9c** to form films on a glass substrate was considered. Gratifyingly, a stable film could be prepared from a CHCl₃ diluted solution of **9c** (15mg/mL), and spin-coated (speed: 2000rpm) on a glass substrate treated by UV/O₃ during 30 minutes prior to use. Pictures of this film are shown below and are now added in the revised ESI version (left normal size from Canon camera EOS 700d, right optical image from a Nikon Eclipse Ti-E microscope).

Both the film thickness and the surface topography were obtained by Surface Force Microscopy in tapping mode (SFM, Bruker Dimension FastScan). Silicon cantilevers (FastScan A) with a tip radius of around 5 nm were used. The resonance frequency of the cantilevers was about 1400 kHz. A 10x10 μm SFM topographic image is shown below, illustrating the homogeneous film formation.

A film thickness of 85 nm was determined by SFM (see right graph below showing surface's profile) *via* a scratch made on the film (see left image below):

Emission of the film obtained from **9c** was determined on Jasco Spectrofluorometer (FP-8300) at excitation wavelength 480 nm (see spectrum below)

It confirms the photoemissive property of this polymer centered at 610 nm.

Comment. “In addition, while the *N*-PPVs do appear to absorb lower energy light (~475 nm) than their analogous dimers (~400 nm) which may reflect an increased conjugation length, do the electronic signatures improve upon known PPVs? For example, MEH-PPV, which is commercially available, absorbs at a similar wavelength at the *N*-PPVs and can be prepared in molecular weights that exceed 50 kDa and thus readily form films. MEH-PPV and other electron rich PPVs are also photoemissive and afford high quantum yields. Additional data and/or clarification are needed. [NOTE: this issue remains unaddressed. The authors are encouraged to provide clear statements that tout the potential advantages of the *N*-PPVs versus known PPVs (e.g., MEH-PPV, etc.) to inform readers of the significance of the work described.]”

Answer. The reviewer is correct regarding the similar absorption property of MEH-PPV and *N*-PPV. We would like to emphasize again, however, the crucial role of the two amino groups in α -position of C=C bonds. In contrast to classical PPV's (including MEH-PPV) and poly(tetra-aminoalkene)s, the presence of two basic nitrogen atoms adjacent to C=C bonds allows the properties of *N*-PPV's to be reversibly modified upon protonation, as detailed in response to comment 4 of reviewer#2. We have also demonstrated that the nature of the amino group has a strong influence on the site of protonation, leading to *N*-PPV's with distinct behaviors: while the conjugation was entirely broken with piperidine substituents (**10b**), it was preserved in the case of di-isopropylamino substituents (**10a**; see Figure 5). Finally, homogeneous films of *N*-PPV's could also be formed despite their low molecular weight (Figure S1-S4); please also refer to the data provided in the response of the 4th comment by reviewer#1. Thus, photoemission of all *N*-PPV's (**10a,b,c**) could be recorded from their respective thin films. Nevertheless, photoemissive properties appeared modest as compared to classical PPV's as quantum yields values could not be obtained due to the rather low intensities. Concerning photoemission of *N*-PPV's in solution, novel Figures, namely, Figures S14, S15, S16, have been included in the ESI. While these works establish the proof of concept for tuning the properties of *N*-PPV's, future work should enable further tuning of those properties, for instance, upon modification of the aryl group or the amino moieties, or by using a variety of alkylating agents for quaternizing the amino groups.

Comment. "Other issues include:

The passage "...reported that peculiar bis-benzimidazol-2-ylidene (sic) can undergo..." should be clarified as ref. 11 reports multiple bis(NHC)s which undergo dimerization. [NOTE: this issue appears to have been addressed.]

The statement "...are also air-sensitive, which makes them not best suited for applications" is misinformed and misleading. Many conjugated polymers (e.g., polyacetylene, polypyrrole, etc.) are air sensitive yet have found utility in a broad range of applications. [NOTE: this issue appears to have been addressed.]"

Answer. As acknowledged by the reviewer, these issues had been already addressed in the version initially submitted.

Reviewer#2

Comment. "The manuscript reports a polymerization through the dimerization of singlet carbene species. The concept is interesting, but I believe the work has not reached the point for publication. In particular, for the very limited scope of polymerization the molecular weights, 3000 g mol⁻¹, 3900 g mol⁻¹ and 5200 g mol⁻¹ for 9a, 9b and 9c, are pretty low and there is no information about polymerization optimization. The molecular weights may be much improved by increasing the concentration of monomers or reaction time, etc."

Answer. Please, see the response given above to reviewer#1, as the same comment was raised.

Comment. "And what is the main side reaction of this polymerization resulting in the low molecular weight?"

Answer. This is an interesting point raised by the reviewer. It can be hypothesized that steric hindrance around the carbene center increases as the chain length increases, i.e. for higher DP. Consequently, dimerization might be kinetically disfavored for DP values reaching 7-10, irrespective of the experimental conditions (nature of the base, concentration, solvent; see also our response to reviewer#1 on this point). We may also propose that a beta-elimination reaction involving the amino-substituent of the carbene center can compete with the dimerization reaction, at least in the case of the iPr₂N derivative. We have therefore performed new DFT calculations regarding the intramolecular beta-elimination of propene from the di-isopropylamino(aryl)carbene **2a**. While this reaction is much less exergonic (-18.5 kcal/mol) compared to the dimerization process (-35.3 and -45 kcal/mol for the dimerization of **2a** into **3a-Z** and **2a** into **3a-E**, respectively), the associated energetic barrier was computed to be only 1-2 kcal/mol higher than that of the dimerization reaction (19.2 kcal/mol vs. 17.2 and 18.5 kcal/mol for **2a**→**3a-Z** and **2a**→**3a-E**, respectively). Obviously, this beta-elimination pathway is expected to be less favorable in the case of the cyclic piperidine substituent. The revised manuscript makes a mention to these new calculations, which have also been added in the revised SI (see Figure S24).

Comment. "Heteroaryl on the main chain plays an important role in the conjugated polymers with optoelectronic properties. I am interested in whether any experiment is done on the acyclic amino(heteroaryl)carbene and its polymerization. The scope of the substances may need to be further broaden."

Answer. The reviewer is right. To broaden the scope of N-PPV's, we targeted an iminium precursor incorporating a thiophene moiety as the heteroaryl substituent, and a piperidine amino group. This compound was prepared in 73% yield following a similar procedure to that used for the other iminium salts prepared in this work. However, deprotonation using LiHMDS or P₄tBu did not lead to a clean reaction, and the corresponding dimer could not be isolated. To investigate the influence of the amino moiety, a phenyl-substituted iminium incorporating a diethylamino or morpholine substituents were also prepared. Unfortunately, deprotonation of these new iminium precursors did not yield a clean reaction either, and no dimer could be isolated. Given those disappointing results

employing mono-functional iminium precursors, the preparation of related bis-iminiums was not attempted. All this information now appears in the revised SI.

Comment. "The presence of amino groups in alpha-position of C=C double bonds in N-PPV's allows their physico-chemical properties to be reversibly manipulated upon chemical post-modification. Whether this phenomenon is just suitable for 9b, or chemical post-modification of the other two polymers, 9a and 9c, leads to the same results. And I am interested in how color of the solution changed during protonation/deprotonation."

Answer. As suggested by the reviewer, protonation of **9a** and **9c** was investigated under similar conditions to those described for **9b**. In particular, a 50:50 mixture of CHCl₃ and CH₃CN was used to accommodate both the solubility of the parent N-PPV **9**, which is a neutral polymer, and the solubility of the as-protonated N-PPV **10**. All solutions of neutral N-PPV's **9a**, **9b** and **9c** display a deep red color. While a similar behavior was observed during protonation of **9b** and **9c** with 2 equivalents of triflic acid (TfOH), a different scenario was observed in the case of **9a**, on the basis of UV/Visible analysis. The solution turned indeed yellow upon protonation of N-PPV's **9b** and **9c**, which translated into a complete loss of absorption at 476 and 483 nm in their respective UV/vis spectrum. In contrast, protonation of **9a** afforded an orange-brownish solution, where the initial absorption at 462 nm shifted to 352 nm. Thus, while the conjugation seemed completely broken in the case of **10b** and **10c**, the situation appeared to be different for **10a**. We hypothesize that these differences are due to the nature of the amino group connected to the C=C bond. Indeed, depending on the basicity and the steric bulk around the nitrogen atom, protonation may either occur at the nitrogen atom or at the carbon atom of the enamine moiety (see Scheme below).

Thus, we wanted to get more insight into the structure of the protonated N-PPV **10**, and to rationalize the difference between the reactivity of di-isopropylamino-N-PPV **9a** and piperidino-N-PPV's **9b**, **c** towards TfOH. For this purpose, protonation of the corresponding dimers, **3a** and **3b** (as representative of **3c**), was investigated with 2 equivalents of TfOH (see scheme below). Protonation of **3a-E/Z** and **3b-E/Z** yielded a mixture of mono-protonated Z-diaminoalkene and di-protonated E-diaminoalkene in both cases. While the structure of the mono-protonated Z-diaminoalkene **Z-3a,b-HOTf** proved to be the same, the structure of the dicationic products were totally different, according to X-Ray analysis. In the case of **3a** featuring two bulky iPr₂N groups, protonation indeed occurred at each nitrogen atom, leading to **E-3a-(HOTf)₂**, where the C=C bond was preserved. In sharp contrast, one nitrogen atom was protonated together with the enamine carbon in the case of **3b** featuring piperidine substituents, yielding **E-3b-(HOTf)₂** without any C=C bond forming. These new results regarding the chemical post-modification of amino-containing dimers and polymers, as well as the corresponding X-Ray structures, now appear in the revised manuscript.

Other points. "(1) The size exclusion chromatography spectra described in Fig. 2 (D) with poor resolution, which can not attain publishable standard. Please revise this point. Figure S10 is with the same problem."

Answer. The resolution of the SEC trace has now been improved in Figure 2D. We are not sure about Figure S10 as this Figure does not exist.

“(2) There are some mistakes in Fig. 4.

a. The color of the chemical structure of a, b, c dose not match the line color of 3a, 3b, 3c and 9a, 9b, 9c.

b. Vertical coordinate of Fig. 4 (B) is not in accordance with that of Fig. 4 (A).

c. The end of UV spectra of polymers should infinitely approach zero like that of Fig. 4 (B).

d. In the chemical equation of Fig. 4 (B), the reaction conditions are not in agreement with the description in page 6. Please check it again.”

Answer. We thank the reviewer for his careful reading. All mistakes in Figure 4 have been corrected in the revised version.

“(3) Some attention needs to be paid to the References Section. Upon looking up the references, I noticed format of some references are wrong.

a. The hyphens, like in Re 8, 23, 25 etc, should be corrected.

b. In Ref 33, ‘Polym. Rev ’ should be ‘Polym. Rev. ’.”

Answer. The reference section has been carefully revised and all typos have been corrected.

(4) In the Materials and Methods Section, format of describing reactions should be paid much more attention, such as ‘3x5mL’, ‘0.048g’, ‘78°C’.

Answer. The “Materials and Method Section” has been updated by correcting and homogenizing format of describing reactions as recommended by the referee.

(5) Emission spectrum of compounds 9b (Figure S8) in thin film is obviously unnormal, the end of the spectrum is almost vertical to the coordinate. It is necessary to collect this emission spectrum again or demonstrate this phenomenon. What’ more, why are the data of emission spectra of polymers in solution lacking?

Answer. The emission spectrum of compound **9b** (Figure S18) has been recorded on a novel sample; however, we observed the same phenomenon, which might be attributed to some inhomogeneity in the film.

Reviewer#3

Comment. *“My major concerns with this manuscript is ther stability of the obtained dimers and polymers. Dimerization of two stable singlet carbenes could lead to trans-bent double bonds (refs 19, 21, 22) and these are known not to be very stable (see ref 22 for an equilibrium between NHC and its enteramine).If the authors can satisfactorily address the question of stability of their dimers and polymers, I believe that the manuscript meets the standards of Nat. Commun. In addition, some typos (bold face for compounds in Figure 3a and in other graphics) should be corrected. Finally, ref 47 should be changed to state correctly that the asymmetric unit of 3a-E contains two independent molecules.”*

Answer. We thank the reviewer for these comments.

Firstly, regarding the formation of trans-bent double bonds, we would like to emphasize that such double bonds have been predicted for heavier analogs of ethylene only, for instance for Si₂H₄, Ge₂H₄ or Sn₂H₄. Indeed, trans-bent distortion of the double bond is observed when $\sum\Delta E_{ST} \geq 0.5 E_{\sigma+\pi}$, where ΔE_{ST} is the singlet-triplet gap of the carbenoid, and $E_{\sigma+\pi}$ the energy of the double bond. In the case of the di-isopropylamino(aryl)carbene studied in this work, DFT calculation gave a ΔE_{ST} of about 20 kcal/mol; the energy of a C=C bond in ethylene is about 172 kcal/mol. Hence, $\sum\Delta E_{ST} = 40$ kcal/mol and $0.5 E_{\sigma+\pi} = 66$ kcal/mol. Therefore, the resulting diaminoalkene is not trans-bent.

To address the important question of the stability of such dimer, the energy of the central C=C bond can be approximated using the following equation: $EC=C(\text{dimer}) = EC=C(\text{CH}_2\text{CH}_2) - 2 \Delta E_{ST} = 172 -$

$2 \times 20 = 132$ kcal/mol. This is about 1.5 higher than the energy of a C-C single bond. This indicates that the dimer is not prone to dissociation. In addition, and from an experimental viewpoint, the stability of dimer **3a** was studied in toluene- d_8 , by monitoring its evolution by ^1H NMR as a function of the temperature. No sign of carbene formation, nor evolution into a decomposition product from the carbene, could be detected over the range of temperature investigated (from 20 to 90 °C). A superimposition of the different ^1H NMR spectra as function of temperature (Figure S8) is given hereafter (full spectra and partial spectra of selected areas). These data further highlight the stability of such dimer.

The corresponding polymer was not probed by temperature ^1H NMR spectroscopy, simply because the broad signals of *N*-PPV's are poorly informative. However, a similar stability of the C=C bond as that observed for **3a** is expected. We can also point out that *N*-PPV's perfectly withstand a temperature of 80 °C for several hours during the Soxhlet extraction in chloroform implemented to purify these polymers. SEC traces remained the same indeed, before and after workup.

Finally, footnote 47 (now footnote 49 in the revised manuscript) was changed to correctly state that the asymmetric unit of **3a-E** contains two independent molecules.

REVIEWER COMMENTS

Reviewer #1 (Remarks to the Author):

While the manuscript continues to see improvement, it is not ready for publication because it contains numerous statements and conclusions that remain unaddressed, are untrue, or are undermined by literature precedent.

(1) The abstract states, "Despite the ubiquity of singlet carbenes in chemistry, their utility as true monomeric building blocks for the synthesis of functional organic polymers has been overlooked." The utility has not been "overlooked". As documented in the references, there are a multitude of publications and reviews on the topic. For example, Ref. 29 (Chem. Commun. 2006, 1727) shows how singlet carbenes can be generated in situ and used to synthesize organic polymers. Similar approaches have been used to prepare organometallic polymers that feature NHCs. The statement needs to be removed.

(2) While it is appreciated that the authors made attempts to increase the molecular weights of the polymers, the persistent small sizes are concerning. Are the newly-described films free-standing or self-supporting? They do not appear to be. The authors may be confused on this point so allow me to explain. Chain entanglement occurs once polymers reach a certain molecular weight. This feature enhances the mechanical properties of the polymers and enables them to become free-standing or self-supporting (i.e., the film can be removed from the substrate and stretched without detriment). Such a characteristic is important for many conjugated polymers, particularly PPV, as it has been shown that the stretching ratio has a profound effect on the conductive and other electronic properties of the material (see: Synth. Met. 1997, 89, 11). Since the manuscript repeatedly makes comparisons to PPV, it behooves the authors to place such comparisons on equal footing. While emphasis was placed on absorption characteristics, the mechanical properties (i.e., film forming ability) are equally important. If sufficiently high molecular weights cannot be achieved, then the value of the polymers will not be realized.

(3) While adding triethylamine may have afforded "better" [response letter] SEC data, the observation is empirical and should be explained. The authors are also encouraged to use a separate technique (e.g., MALDI MS) to corroborate molecular weight data as the amine additive may merely be modulating the size exclusion effects of the chromatography columns used and not providing a more accurate description of polymer molecular weight.

(4) The issue of whether or not the electronic signature of N-PPV is an improvement upon known PPVs remains unaddressed. I understand that the N-PPV can be treated with an acid which, in turn, changes its absorption profile, but how does this improve upon known PPVs? The authors tout the "switchable 'on-off' absorption" characteristic of the N-PPVs as an advantage, but please realize that PPVs outfitted with basic groups are known and have been reported "to exhibit reversible and tunable optical properties depending on protonation-deprotonation processes"; see: J. Am. Chem. Soc. 1998, 120, 10463.

Overall, the work described is fascinating but still oversold and underdelivered. However, with additional clarification and context placement, I am of the opinion that the paper can evolve into a form that will be appreciated by a niche audience.

Reviewer #2 (Remarks to the Author):

The authors have properly addressed the reviewers' concern and revised their manuscript accordingly. The revised manuscript has been significantly improved. I recommend the acceptance of this work for publication in Nature Communications.

Reviewer #3 (Remarks to the Author):

An edited version of manuscript NCOMMS-20-32100 has been reviewed. The points I have raised in my previous review regarding the stability of carbene dimers and polymers have been addressed satisfactorily. While the topics raised by the other two reviewers retain their validity, I have no further remarks and support publication of the manuscript in its present form.

Direct and Selective Access to Amino-Poly(Phenylene vinylene)s with Switchable Properties by Dimerizing Polymerization of Aminoaryl Carbenes

Manuscript NCOMMS-20-32100: Answers to reviewer#1's comments

General comment. *"While the manuscript continues to see improvement, it is not ready for publication because it contains numerous statements and conclusions that remain unaddressed, are untrue, or are undermined by literature precedent."*

Answer. We understand the feeling of the reviewer, even if we do not fully share the same opinion as him/her on some of the points raised. We sincerely thank her/him for these exchanges, which in turn allow us to improve the manuscript even more. Hereafter, we provide a detailed point-by-point response to her/his remarks and questions.

Comment. *The abstract states, "Despite the ubiquity of singlet carbenes in chemistry, their utility as true monomeric building blocks for the synthesis of functional organic polymers has been overlooked." The utility has not been "overlooked". As documented in the references, there are a multitude of publications and reviews on the topic. For example, Ref. 29 (Chem. Commun. 2006, 1727) shows how singlet carbenes can be generated in situ and used to synthesize organic polymers. Similar approaches have been used to prepare organometallic polymers that feature NHCs. The statement needs to be removed.*

Answer. The revised manuscript already mentioned the state of the art regarding the use of bis-carbenes, in the form of their precursors, as monomeric building blocks for polymer construction. New references had been added to account for this background literature in the former revised version of the manuscript. To take into account the reviewer's remark on that point, the term "overlooked" has been changed into "underexplored" in the first sentence of the abstract.

Comment. *While it is appreciated that the authors made attempts to increase the molecular weights of the polymers, the persistent small sizes are concerning. Are the newly-described films free-standing or self-supporting? They do not appear to be. The authors may be confused on this point so allow me to explain. Chain entanglement occurs once polymers reach a certain molecular weight. This feature enhances the mechanical properties of the polymers and enables them to become free-standing or self-supporting (i.e., the film can be removed from the substrate and stretched without detriment). Such a characteristic is important for many conjugated polymers, particularly PPV, as it has been shown that the stretching ratio has a profound effect on the conductive and other electronic properties of the material (see: Synth. Met. 1997, 89, 11). Since the manuscript repeatedly makes comparisons to PPV, it behooves the authors to place such comparisons on equal footing. While emphasis was placed on absorption characteristics, the mechanical properties (i.e., film forming ability) are equally important. If sufficiently high molecular weights cannot be achieved, then the value of the polymers will not be realized.*

Answer. Investigation into the film forming properties of *N*-PPV's was performed during the first round of revisions, following the initial comments of the reviewer. And study of the mechanical properties of our polymers was neither required nor mentioned by the reviewer at that time. We obviously acknowledge that this is a very important feature in the perspective of possible applications of our materials. However, we think that such investigations are out of the scope of this work. From the set of data presented in the revised version, films we have made, despite the modest molar mass of *N*-PPV's, proved to be self-supported, as are the organic materials classically used in optoelectronics. It is noteworthy that conjugated polymers employed in these applications are most often integrated as thin films on rigid substrates within a multi-layer stack through various printing techniques. For instance, here we have used standard spin-coating technique and orthogonal solvents to form films from our *N*-PPV's.

Comment. *While adding triethylamine may have afforded "better" [response letter] SEC data, the observation is empirical and should be explained. The authors are also encouraged to use a separate*

technique (e.g., MALDI MS) to corroborate molecular weight data as the amine additive may merely be modulating the size exclusion effects of the chromatography columns used and not providing a more accurate description of polymer molecular weight.

Answer. As stated by this referee, this observation is rather empirical. In general, adding a solute showing similar properties to that of a polymer analyzed by SEC can decrease inter-chain polymer interactions, as well as interactions with the stationary phase. Thus, amines, such as triethylamine, can be added to analyze amino-containing polymers, while small amounts of acids can be used for acidic-containing polymers. Addition of triethylamine to characterize amino-containing polymers by SEC has been described in many publications. Three textbook references have been to address the reviewer's comment: 1) added (1. Mori, S & Barth, H (1999) *Size Exclusion Chromatography*, Springer Verlag, Berlin, Germany; 2) Wu, C-S (2003) *Handbook of Size Exclusion Chromatography and Related Techniques*. Marcel Dekker, New York, NY, U.S.A.; 3) Striegel, AM, Yau, WW, Kirkland, JJ & Bly, DD (2009) *Modern Size-Exclusion Chromatography*. John Wiley & Sons, Chichester, UK).

As also suggested by the reviewer, MALDI ToF analysis was performed on 2 samples, namely, *i*Pr₂N-PPV **9a** and the fluorene derivative PipN-PFV **9c**. Corresponding MALDI ToF mass spectra are shown in the revised ESI (see Figures S20 and S21). In both cases, we were pleased to be capable to characterize the targeted polymers with *m/z* up to 1719 and 4635, for **9a** and **9c**, respectively, showing an aldehyde chain end at both sides. As expected, isotopic peaks are separated by *m/z* = 300 and 524 in the case of **9a** and **9c**, respectively, corresponding to the molar mass of the expected monomer unit. Furthermore, a perfect agreement was noted between the experimental and simulated isotopic pattern for a selected *m/z* value, validating the proposed structure for **9a,c**. Unexpectedly, each of these *N*-PPV's revealed the presence of a structure incorporating both an iminium and an aldehyde chain ends. We hypothesized that this population resulted from an incomplete deprotonation of the bis-iminium precursor. Yet, C-H iminium signals could not be detected by ¹H NMR, likely due to their presence in low concentration. Interestingly, these cationic type polymer chains can be well detected by MALDI ToF MS under our experimental conditions. Finally, a cyclic conjugated polymer structure incorporating 3 to 9 repeating units could also be identified in the case of compound **9c**.

Comment. *The issue of whether or not the electronic signature of N-PPV is an improvement upon known PPVs remains unaddressed. I understand that the N-PPV can be treated with an acid which, in turn, changes its absorption profile, but how does this improve upon known PPVs? The authors tout the "switchable 'on-off' absorption" characteristic of the N-PPVs as an advantage, but please realize that PPVs outfitted with basic groups are known and have been reported "to exhibit reversible and tunable optical properties depending on protonation-deprotonation processes"; see: J. Am. Chem. Soc. 1998, 120, 10463.*

Answer. As mentioned in our former exchanges with the reviewer, *N*-PPV's have similar absorption profile than regular, namely, nonamino-substituted PPV's (e.g. MEH-PPV's). In contrast to these PPV's and poly(tetra-aminoalkene)s, however, the presence of the two basic nitrogen atoms adjacent to C=C bonds allows the properties of *N*-PPV's to be reversibly modified. We have also established that the nature of the amino group has a dramatic impact on the site of protonation, leading to *N*-PPV's with distinct behaviors. Thus, while conjugation is entirely broken after protonation of compounds with piperidine substituents (**10b**), protonated di-isopropylamino-containing polymers (**10a**; see Figure 5) remain conjugated, highlighting the crucial role of the two amino groups in α -position of C=C bonds. We thank the reviewer for bringing up the reference concerning the switchable properties of PPV's (*J. Am. Chem. Soc.* **1998**, *120*, 10463). Such PPV's, specifically incorporating bipyridylene-vinylene subunits, were synthesized by repeated Wittig olefination reactions. Nitrogen atoms of those polymers are therefore not directly connected to the C=C double bond, as in the case of our *N*-PPV's. Though we acknowledge that they can be reversibly protonated, this yields to a red-shift of the maximum of absorption for the corresponding cationic polymer. In sharp contrast, the protonation of the amino-groups in our *N*-PPV's leads to a blue-shift or to a disappearance of the maximum of absorption, depending on the nature of the amino-groups. Again, this original behavior results from the particular

α -positioning of these amino groups in *N*-PPV's, which to the best of our knowledge, had never been observed before, neither in traditional PPV's nor in PPV containing bipyridylene-vinylene subunits. Thus, we do not wish to emphasize about "improved properties" of *N*-PPV's, but rather point out their specific and original properties, compared to that of classical PPV-like materials. In order to highlight this difference, the following sentence has been added into the abstract (in yellow):

"Hence, depending on the nature of the amino group (iPr₂N vs. piperidine), either a complete loss of conjugation or a blue-shift of the maximum of absorption is observed, as a result of the protonation at different sites (nitrogen vs. carbon)".

The following sentence has also been modified in the introduction: *"These properties may even be finely tuned upon post-chemical modification of the basic amino groups... in contrast with the behavior of PPVs without alpha-amino substituents"* (in yellow). This may notably be of interest to the preparation of sensor-type materials.

The reference mentioned by the reviewer has been added (now reference 44 of the revised manuscript), as follows: *"These properties may even be finely tuned upon post-chemical modification of the basic amino groups in 1,2-position of the double bonds"*.

REVIEWERS' COMMENTS

Reviewer #1 (Remarks to the Author):

These repeated exchanges are futile. As I have stated repeatedly: I like this paper and am of the opinion that the work should be published, but in a specialized journal. Beyond the lack of a proper comparison to commercially available PPVs, the difference between the current work and that presented in JACS 1998, 10463 is nuanced. The authors acknowledge this: "[The] [n]itrogen atoms of those polymers are ... not directly connected to the C=C double bond, as in the case of our N-PPV's. Though we acknowledge that they can be reversibly protonated, this yields to a red-shift [whereas] the protonation of ... our N-PPV's leads to a blue-shift". The incremental refinement will be appreciated by the community interested in refined PPVs; however, the general audience has been informed of the key advance for more than 20 years.

Reviewer #2 (Remarks to the Author):

The authors have properly addressed the reviewer's concern and revised their manuscript properly. I support the publication in its current form.

Direct and Selective Access to Amino-Poly(Phenylene vinylenes)s with Switchable Properties by Dimerizing Polymerization of Aminoaryl Carbenes

Manuscript NCOMMS-20-32100: Answers to reviewers' comments -
May 22nd, 2021

Reviewer#1

General comment. These repeated exchanges are futile. As I have stated repeatedly: I like this paper and am of the opinion that the work should be published, but in a specialized journal. Beyond the lack of a proper comparison to commercially available PPVs, the difference between the current work and that presented in JACS 1998, 10463 is nuanced. The authors acknowledge this: "[The] [n]itrogen atoms of those polymers are ... not directly connected to the C=C double bond, as in the case of our N-PPV's. Though we acknowledge that they can be reversibly protonated, this yields to a red-shift [whereas] the protonation of ... our N-PPV's leads to a blue-shift". The incremental refinement will be appreciated by the community interested in refined PPVs; however, the general audience has been informed of the key advance for more than 20 years.

Answer. In spite of the doubts again expressed by reviewer#1, we believe that this work innovates the field of the synthesis of π -conjugated polymers. Notoriously, it also widens the field of applications of singlet carbenes through their use as building blocks for polymer synthesis. We thus expect that our results will have an important echo and a great visibility, when they are published in *Nature Communications*.

* * * * *